# Influence of Seed Source and Soil Contamination on Ecophysiological Responses of *Lavandula pedunculata* in Rehabilitation of Mining Areas

**DOI:** 10.3390/plants11010105

**Published:** 2021-12-30

**Authors:** Daniel Arenas-Lago, Luisa C. Carvalho, Erika S. Santos, Maria Manuela Abreu

**Affiliations:** 1Department of Plant Biology and Soil Science, University of Vigo, As Lagoas, Marcosende, 36310 Vigo, Spain; 2Área de Edafoloxía e Química Agrícola, Facultade de Ciencias, Universidade de Vigo, As Lagoas, 32004 Ourense, Spain; 3Linking Landscape, Environment, Agriculture and Food Research Centre (LEAF), Associated Laboratory, Laboratory for Sustainable Land Use and Ecosystem Services (TERRA), Instituto Superior de Agronomia, Universidade de Lisboa, 1349-017 Lisboa, Portugal; lcarvalho@isa.ulisboa.pt (L.C.C.); erikasantos@isa.ulisboa.pt (E.S.S.); manuelaabreu@isa.ulisboa.pt (M.M.A.)

**Keywords:** Iberian Pyrite Belt, contamination, potentially hazardous elements, environmental rehabilitation, oxidative stress, antioxidative activity

## Abstract

Mining activities have turned many areas of the Iberian Pyrite Belt (IPB) into extreme environments with high concentrations of metal(loid)s. These harsh conditions can inhibit or reduce the colonization and/or development of most vegetation. However, some species or populations have developed ecophysiological responses to tolerate stress factors and contaminated soils. The main objectives of this study are: (i) to assess the differences in germination, growth, development and physiological behaviour against oxidative stress caused by metal(loid)s in *Lavandula pedunculata* (Mill.) Cav. from two different origins (a contaminated area in São Domingos mine, SE of Portugal and an uncontaminated area from Serra do Caldeirão, S of Portugal) under controlled conditions; and (ii) to assess whether it is possible to use this species for the rehabilitation of mine areas of the IPB. After germination, seedlings from São Domingos (LC) and Caldeirão (L) were planted in pots with a contaminated soil developed on *gossan* (CS) and in pots with an uncontaminated soil (US) under controlled conditions. Multielemental concentrations were determined in soils (total and available fractions) and plants (shoots and roots). Germination rate, shoot height, dry biomass and leaf area were determined, and pigments, glutathione, ascorbate and H_2_O_2_ contents were measured in plant shoots. Total concentrations of As, Cr, Cu, Pb and Sb in CS, and As in US exceed the intervention and maximum limits for ecosystem protection and human health. The main results showed that *L. pedunculata*, regardless of the seed origin, activated defence mechanisms against oxidative stress caused by high concentrations of metal(loid)s. Plants grown from seeds of both origins increased the production of AsA to preserve its reduction levels and kept the contents of GSH stable to maintain the cell’s redox state. Plants grown from seeds collected in non-contaminated areas showed a high capacity for adaptation to extreme conditions. This species showed a greater growth capacity when seeds from a contaminated area were sown in uncontaminated soils. Thus, *L. pedunculata*, mainly grown from seeds from contaminated areas, may be used in phytostabilization programmes in areas with soils with high contents of metal(loid)s.

## 1. Introduction

Ore and coal mining is one of the anthropic activities that causes great ecosystem disturbance and contamination problems, due to the release into the environment of high contents of metal(loid)s as a consequence of, directly or indirectly, contaminated and reactive tailings [1,2]. In general, unfavourable physical and/or chemical characteristics of soils, developed on mine wastes and/or natural soils influenced by adjacent tailings and/or acid mine drainage, reduce or inhibit the colonization and development of most plant species. In fact, the presence of high contents of metal(loids) in the soils/mine wastes is one of the abiotic factors that can lead to major problems of oxidative stress in most plants. These plants suffer an increment and accumulation of reactive oxygen species (ROS) in their cells and, consequently, considerable damage at the physiological and cellular levels [3,4].

Nonetheless, there are some species (e.g., *Lavandula* and *Cistus* genus) and/or populations able to survive and develop all their life cycle in both contaminated and uncontaminated areas, which can be considered species with the capacity to recover contaminated soils and improve the edaphic conditions of these mining areas [2,5,6,7,8,9]. This adaptation is due to the decrease in uptake, translocation and/or accumulation of potentially toxic elements in photosynthetic active parts as well as to the stimulation of several physiological mechanisms to scavenge ROS and overcome oxidative stress [2,5,6,7,8,9]. These physiological mechanisms in plants growing in the presence of metal(loids), in particular in mining areas, can include an increase in antioxidant enzyme activities (e.g., catalase, peroxidase), as well as non-enzymatic antioxidants such as ascorbic acid (AsA), reduced glutathione (GSH), α-tocopherol, carotenoids, anthocyanins and flavonoids, among others [2,6,7,10].

The growth of tolerant species improves the physical, chemical and biological properties of soils developed on mine wastes or even mine wastes with multielemental contamination, increasing the vegetation cover and contributing to the environmental rehabilitation of these degraded areas [11]. Several biogeochemical studies have been carried out in different Iberian Pyrite Belt (IPB) mines in order to identify species growing spontaneously and to assess their potential for environmental rehabilitation through phytostabilization (e.g., [5,6,8,9,10,12,13]).

Among these species with phytostabilizing potential is *Lavandula pedunculata* (Mill.) Cav. (Synonym: *Lavandula sampaiana* (Rozeira) Rivas Mart., T.E. Díaz & Fern. Gonz and *Lavandula stoechas* L. subsp. *pedunculata* (Mill.) Samp. Ex Rozeira). This species grows spontaneously in uncontaminated soils, forming communities in semi-arid regions in the southwest of the Iberian Peninsula [2,14,15], but it also grows in contaminated soils from mining areas of the IPB [2,16,17,18,19]. In previous studies, this species has been recognized for its phytostabilization potential in mining areas under Mediterranean conditions, as well as for its high economic value for several industries because of the properties of its essential oils [2,19].

In the recovery of the abandoned mining areas demanded by the European Union (Directive 2006/21/EC, 2006; European Commission, 2017), the use of these autochthones and tolerant plants in environmental rehabilitation programmes is an advantage. These plants are well adapted to coexisting with various stress conditions, but they also maintain an ecological succession that is similar to the adjacent natural ecosystem. However, for the implementation of environmental rehabilitation programmes, it is necessary to obtain a considerable number of seeds, preferably from plants growing spontaneously in contaminated areas. This task is usually very difficult since plant populations are generally small and have a low number of individuals so the use of seeds from plants growing in non-contaminated sites, such as nurseries, can be a solution. Thus, it is necessary to understand the physiological responses of the different ecotypes, since plant seeds from contaminated and non-contaminated areas can present contrasting behaviours and contribute differently to soil rehabilitation.

Currently, the main studies on the applicability of *L. pedunculata* are largely focused on characterising and evaluating the bioactivity of essential oils and other extracts of this species with antioxidant, antifungal and antibacterial properties [17,20,21,22,23,24], although its potential as a species to recover contaminated areas must also be considered. *Lavandula pedunculata* shows high levels of free-radical scavengers and therefore it can be a potential source of active metabolites with a positive effect on human health. Hence, the antioxidant properties of this species might also be an economic asset for populations if it was used for soil recovery in contaminated areas [2,19]. Therefore, the main objectives of this research were (i) to assess the possible differences in germination, growth, development and physiological behaviour against oxidative stress caused by metal(loid)s in *L. pedunculata* from two different origins (a contaminated mining area in the IPB and an uncontaminated area) under controlled conditions (greenhouse); and (ii) to assess whether it is possible to use this species for the rehabilitation of the IPB mine areas.

## 2. Results

### 2.1. Soil Characterization

Physico-chemical characteristics of the contaminated (CS) and uncontaminated (US) soils are shown in Table 1. The CS (collected in São Domingos mining area) presents acid pH, very low concentrations of organic carbon, total N and extractable P but very high concentration of extractable K. The low N contents and the very high C/N ratio indicate that the organic matter decomposition is very slow. The US (collected in Serra do Caldeirão) also shows very low concentrations of organic C, total N and extractable P. However, pH values are moderately acidic and extractable K concentration is high. In general, the fertility in US can be considered lower than in CS. According to different reference guidelines for metal(loid) concentrations in soils [25], the total concentrations of As, Cr, Cu, Pb and Sb in CS, and As in US exceed the intervention and maximum permitted levels for the protection of ecosystems and human health, specifically for industrial land use. In both soils, the elements’ concentrations in the soil available fraction were low when compared to their total concentrations, although Cr, Fe, K, Mg and Ni were slightly higher in US than in CS.

### 2.2. Germination and Growth of L. pedunculata

#### 2.2.1. Germination Assays

Germination over time and germination rates of *L. pedunculata* seeds (LC and L) in the soils (US and CS) are shown in Figure 1. Seeds from Serra do Caldeirão (L) had a high and quick germination capability. In fact, approximately eight days after sowing, *L. pedunculata* seeds from Caldeirão had already germinated independently of the soil (US-L and CS-L), but *L. pedunculata* seeds from São Domingos did not germinate in any soil (US-LC and CS-LC) (Figure 1). Over time, the germination of LC and L in both soils increased progressively, with the number of germinated seeds from Caldeirão (L) always higher than that of seeds from São Domingos (LC). However, at the end of the assay (after 30 days), the germination rates only showed significant differences among CS-LC (the lowest germination values) and US-LC and US-L.

#### 2.2.2. Plant Development

Parameters (shoot height, dry biomass, leaf area and leaf area ratio) associated to plant development are presented in Figure 2. Shoot height increased progressively (for 35 days) in all soil-plant sets (Figure 2). Specifically, soil-plant sets showed no significant differences among shoot height during the first week after transplantation. In addition, in the following weeks, no differences were obtained in the height of the different plants (LC or L) in the same soil. Plants from seeds collected in Caldeirão (L) had similar heights independently of the soil but LC plants presented higher development in US than CS.

The amount of shoot biomass produced by LC growing in US was significantly higher than the other soil-plant sets (Figure 2B). In CS, dry biomasses of shoots and roots were similar for both seed origins. On the other hand, L root biomass production in US reached the lowest values. The ratios of root/shoot dry biomass (CS-LC: 0.47 ± 0.14; CS-L: 0.49 ± 0.14; US-LC: 0.26 ± 0.08; US-L: 0.19 ± 0.05) were higher in CS than US but similar between different seed origins in the same soil.

For leaf area, no significant differences were observed among soil-plant sets (Figure 2C). Although the leaf area was not statistically different between the sets, the lowest leaf area ratio (LAR, ratio: leaf area/dry leaf mass) in US-LC indicated a slightly higher leaf biomass than in the other plant sets (Figure 2D).

In general, plants from LC seeds had better development in uncontaminated soils (US) than CS (and even compared to L in US). Moreover, the characteristics of CS did not negatively affect the plant development suggesting a high tolerance of the species regardless of the origin of the seed population.

### 2.3. Metal(loid)s Concentration in Roots and Shoots

The concentrations of metal(loid)s in the roots and shoots of *L. pedunculata* from different origins (LC and L) growing in both soils (CS and US) are shown in Table 2. In general, there were significant differences for some metal(loid) concentrations in roots and shoots among the different soil-plant sets (Table 2). More specifically, the concentrations of As, Cd, Cr, Cu, Mn, Mo, Pb and Zn were higher in the roots of LC and L in CS than in US (no significant differences for Cu, Mn and Zn in L; Cu and Cr in LC) (Table 2). In shoots, no significant differences were found for Cd, Cr, Mo (except L in CS) and Mn concentrations between L and LC for both soils. However, Pb and As concentrations were significantly higher in CS than in US independently of seed origin. Conversely, Zn and Cu concentrations in shoots were significantly higher in US than in CS independently of seed origin (Table 2). Concerning other studied elements, Ni concentrations (in roots and shoots) were very low in plants growing in CS compared to plants growing in US, and much lower in L than LC, whereas Fe concentrations (in roots and shoots) were higher in plants growing in CS than in US. Calcium, Na and Mg contents in roots were higher in CS than in US, while K was lower.

The translocation coefficient (TranslC) and soil-plant transfer coefficient (TransferC) of metal(loid)s in *L. pedunculata* are shown in Table 3. In contaminated soil (CS), metal(loid)s were, generally, stored in plant roots (TranslC < 1). The exception was found for elements such as Mn and K, which were mainly stored in shoots (Table 2 and Table 3). *Lavandula pedunculata* behaved like an excluder/non-accumulator species (TransferC < 1) for As, Cu, Fe, Mo (in L) and Ni (in L).

In uncontaminated soil (US), macro- and micro-nutrients, such as Ca, Mg, K, Mn and Zn, were translocated from roots to shoots. However, As, Cu, Fe and Ni were mainly stored in roots. A translocation coefficient was not calculated for Cd, Cr, Mo (except in CS-L), Pb (in US), Ni (in CS) and Sb because their shoot and root concentrations were under the detection limit (Table 2 and Table 3). *Lavandula pedunculata* behaved like an excluder/non-accumulator species (TransferC < 1) for As, Cu, Fe, Mg (except in plants from L seeds), Na and Ni. However, Ca, K, Mg (except in LC), Mn and Zn tend to accumulate in shoots (Table 3).

### 2.4. Physiologcal Parameters

#### 2.4.1. Pigment Concentrations and Visual Symptoms

Pigment concentrations in leaves of *L. pedunculata* (L and LC) growing in both soils (US and CS) are shown in Figure 3. Chlorophyll (*a*, *b* and total), anthocyanin and carotenoid concentrations did not show significant differences in any soil-plant set, with the exception of L in US that had the highest chl *b* concentration. The high variability of the pigment contents was consistent with the physiological characteristics that *L. pedunculata* leaves presented at the end of the experiment (Figure 4). Most of the leaves of LC and L in CS showed a normal dark green colour without apparent symptoms of chlorosis. Furthermore, the stems were dark brownish and greenish as the plant height increased. Likewise, most of the leaves of LC and L in US also did not show symptoms of chlorosis, although a few of them had rolled leaves with black and white dots and the abaxial side reddish, mainly those placed in the first levels of the stem. Visual differences in shoot biomass contents were also observed between plants growing in US and CS (Figure 4).

#### 2.4.2. Oxidative Stress and Physiological Response

The H_2_O_2_ contents in leaves of *L. pedunculata* varied considerably depending on the soil-plant set (Figure 5). The H_2_O_2_ contents in leaves from L and LC growing in US were lower than the detection limit (<0.04 μmol g^−1^ FW). However, H_2_O_2_ contents in leaves from L and LC growing in CS were significantly higher than in US.

The ascorbate and glutathione contents in leaves of *L. pedunculata* are shown in Figure 6. There were no significant differences for total ascorbate contents in L and LC in each soil, although there were significantly higher levels of total ascorbate in L and LC in CS compared to the plants from US (>3-fold higher) (Figure 6A). Furthermore, the percentages of reduced ascorbate (AsA) were much lower in L and LC growing in CS than in US (Figure 6A). Total glutathione contents were significantly lower in LC than in L in plants from US. These differences can be explained by changes in GSH levels, with L in US showing the highest levels and LC in US the lowest. This also led to the highest reduction state of glutathione (GSH) in US-L and the lowest in US-LC (Figure 6B).

## 3. Discussion

*Lavandula pedunculata* (LC and L), regardless of the origin of the seeds, was able to grow and produced considerable biomass in US, while in CS grown plants showed some visual symptoms of toxicity. The LC plants showed the greatest development (higher height, more biomass and lower LAR) in US compared to the other plant sets. These differences seem to indicate that seeds from plants adapted to stress conditions produce plants that take advantage of soils with better properties for plant development and the stress conditions decrease. This species, regardless of seed origin (LC or L), grows in CS with an acidic pH (≈3.5) and high total concentrations of metal(loid)s, such as As, Cr, Cu, Pb and Sb, whose concentrations exceed the limits of the intervention and maximum permitted levels [25]. This means that plants obtained from seeds collected in non-contaminated soils (L) have naturally adapted to the stressful conditions, in particular to the high concentrations of potentially toxic elements in CS. Nonetheless, this fact could also be related to the low availability of the potentially hazardous elements in the soils, even in CS (Table 1). The ratio of root/shoot dry biomass can also provide information on the condition of the plants in relation to oxidative stress. Figure 2B shows that there are no significant differences in root/shoot ratios among US-L, CS-L and CS-LC, while this ratio is higher for US-LC. This suggests that *L. pedunculata* from São Domingos can develop better when grown under less oxidative conditions in uncontaminated soils (Figure 4). Likewise, there was no growth impairment in plants obtained from the seeds collected in the uncontaminated area growing in the mine soil (CS-L), compared to plants from seeds of the contaminated area growing in the same soil (CS-LC). This indicates that this species has a high plasticity of adaptation to the oxidative stress conditions, regardless of the origin of its seeds. In fact, the lower growth in LC and L in CS, compared to H_2_O_2_ contents found in individuals growing in US, can be related to the higher contents of H_2_O_2_. Schützendübel and Polle [26] showed that exposure of plants to metals resulted in oxidative stress as indicated by lipid peroxidation, hydroxyl radical production or H_2_O_2_ accumulation.

Several studies have pointed out that different species of genus *Lavandula*, such as *L. dentate* L., *L. luisieri* (Rozeira) Rivas Mart. or *L. stoechas* Lam., are able to develop in soils under high oxidative stress conditions in several areas of the IPB [10,12,27,28]. However, unlike in the present study, these species grew spontaneously in the IPB mine areas and not from seeds harvested from plants growing in uncontaminated areas. Hence, these results support the adaptation and possibility of introduction of *L. pedunculata* from uncontaminated areas into highly contaminated soils from mining areas.

Concerning the accumulation and translocation of metal(oid)s, *L. pedunculata* tends to store these elements in the roots, independently of soil characteristics and seed source, avoiding their translocation to shoots. In the present study, the storage of As, Cu, Pb, Fe and Zn in the roots of plants growing in CS can be considered an efficient defence mechanism to prevent phytotoxic effects [2,29] and may indicate the possible use of this species in phytostabilization programmes [2,30]. Likewise, *L. pedunculata* also acts as an excluder/non-accumulator species for As, Cu and Fe in plants growing in both soils. De la Fuente et al. [18] showed that species of the genus *Lavandula* growing in Rio Tinto, a mine of the IPB, were also excluders of As, Cu and Fe. Other studies, undertaken in species of the *Lamiaceae* family showed storage behaviour of potentially hazardous elements in mine soils of the IPB, similar to our findings [2,27,31].

With a few exceptions, most of the studied element concentrations in the shoots of LC and L species growing in US and CS were considered as normal/sufficient and/or below the phytotoxicity level [32,33]. According to Kabata-Pendias [29], Mn and Zn phytotoxic levels in shoots may be observed above 300–1000 mg kg^−1^. *Lavandula pedunculata* had Zn contents in shoots below these concentrations, although Mn concentrations in shoots were within this range for plants growing in both soils. Arsenic concentrations in shoots of LC and L in CS were above 20 mg kg^−1^, which exceeds the phytotoxic limits for this element (5–20 mg kg^−1^ [29]). Similarly, the onset of Cu phytotoxicity, together with a decrease in yield, was reported in shoots and leaves at concentrations between 5 and 40 mg kg^−1^ [32]. However, *L. pedunculata* had Cu concentrations in this range in shoots of LC and L growing in US, but no Cu toxicity symptoms were observed. Although the shoots of *L. pedunculata* growing in CS had high concentrations of Pb, they were below the values considered excessive in plants grown in mine soils (63–570 mg kg^−1^ depending on the species [29]). In US, *L. pedunculata* (LC and L) did not show any toxicity symptoms. In CS, in general no symptoms of toxicity were found in the majority of the plants. In any case, in a few individuals of LC and L growing in CS some leaves showed chlorotic spots, which can be related with the high concentrations of As found in LC and L shoots. Carvalho et al. [34] described similar damage symptoms in *Cistus monspeliensis* growing in hydroponic solutions with high concentrations of As (>5000 µM). In the current study, no significant differences were found for pigments among soil-plant sets (except for chl *b* contents in US-L). The ratios of chl *a*/chl *b* were high for all soil-plant sets, except for US-L, due to the high chl *b* content. This ratio can be used as an indicator of oxidative stress in plants, due to the fact that metal(loid) accumulation affects each component differently, provoking changes in plant physiology [35]. In fact, the increase of the chlorophyll ratio due to oxidative stress has been reported in *Phaseolus vulgaris L.* exposed to Cd, Cu and Pb contamination [36]. The similar contents of anthocyanins or carotenoids in all plants, regardless of their origin and the soil conditions, was indicative of the stability of these pigments to exposure to high metal(loid) contents. Both pigments act as ROS scavengers and also protect chlorophylls in oxidative stress conditions [37,38].

*Lavandula pedunculata* growing in CS increased the production of ascorbate to preserve some AsA content, since a high reduction rate of ascorbate is necessary to maintain efficient ROS scavenging in cells. Instead, *L. pedunculata* growing in US were able to maintain AsA regeneration and the cell’s redox state with low levels of total ascorbate. In spite of these strategies, AsA (%) was lower in plants of CS than US. This can be explained by the high contents of H_2_O_2_ in LC and L in CS, as a consequence of high oxidative stress in these plants.

The total glutathione contents in LC and L in CS were not significantly different from those of L in US. The reduction ratios were not significantly different, except for LC and L in CS (lower) compared to reduction ratios for L in US. Nevertheless, these values are still within a favourable range to maintain the cell’s redox state. In line with these results, Foyer and Shigeoka [39] indicated that low GSH reduction ratios act as a triggering signal for the activation of the antioxidative stress response. Jozefczak et al. [40] also indicated that GHS acts as a key for metal scavenging due to the high affinity of metals to its thiol (-SH) group. Likewise, Arenas-Lago et al. [41] reported that GHS levels kept high under metal stress conditions may reduce oxidative stress. It must be pointed out that glutathione and ascorbate contents in LC in US were the lowest, indicating that *L. pedunculata* from an extreme environment is able to grow in uncontaminated soil, decreasing its oxidative stress levels and limiting the production of these compounds.

## 4. Material and Methods

### 4.1. Study Areas and Material Sampling

This study was carried out with soils and *Lavandula pedunculata* (Mill.) Cav. seeds collected in two different areas: (i) the São Domingos mining area located in the Portuguese section of the IBP, Alentejo region (SE, Portugal) and (ii) Serra do Caldeirão located in the Algarve region (S, Portugal). In both areas, *L. pedunculata* is a representative species in the vegetation community.

The São Domingos mining area is an old metal-sulfide mine abandoned since 1960, where mining activities began in the pre-Roman period to extract Ag, Au and Cu from *gossan*. Subsequently, in the second half of the nineteenth century and the first half of the twentieth century, massive sulphides, mainly pyrite, were processed to obtain Cu, Zn, Pb and S, among other elements [42]. Mining activities produced large amounts of wastes, which were spread and deposited in several tailings along the area, with different environmental risk according to their volume, chemical and mineralogical characteristics, and potential generation of acid mine drainage [43]. The low pH of the mine wastes and their leachates and the high mobility of the potentially toxic elements (PHE) generated extreme environmental conditions and severe contamination in the entire mine and the surrounding areas, including surface water [43,44]. In fact, soils developed on mine wastes and host rocks (Spolic Technosol; [45]), as well as many adjacent natural soils, which are developed on schists and greywackes (Lithic Leptosols; [45]), present high total concentrations of metal(loid)s [2,8,9,46].

The climate in this area is semi-arid and mesothermic (Thornthwaite classification), with low rainfall (annual precipitation of 548 mm, Climate normal, 1981–2010 from Beja station [47]) and below to the potential evapotranspiration, with warm winters and hot summers within continental climate regimes.

The Serra do Caldeirão is the largest mountain in the Algarve region, which marks the border between the Algarve and Alentejo regions. The lithology of Serra do Caldeirão is mainly composed of schists and greywackes rocks, which originated incipient and shallow soils (Lithic Leptosols and Hyperskeletic/Skeletic Leptosols [45]). It constitutes a physical barrier for cold north-easterly winds, humid southerly winds and north-west depressions, contributing to the existence of a Mediterranean climate with weak annual rainfall around ≈1000 mm and mild winter temperatures due to the altitude [48].

Three composite samples (0–20 cm deep) of a contaminated soil developed on *gossan* from São Domingos mine (CS) and an uncontaminated soil (US) from Serra do Caldeirão were collected. The soils were air-dried, homogenized and sieved (2 mm) before their characterization and use in the assays.

*Lavandula pedunculata* seeds were collected in the same areas (hereafter referred as LC and L for São Domingos and Serra do Caldeirão, respectively) and stored in the dark at room conditions until their use in the assays. These seeds were used for the germination assay and to provide seedlings for the assay to evaluate plant development and ecophysiological parameters.

### 4.2. Experimental Set-Up and Monitoring

In order to evaluate which ecological factor (seeds source or soil conditions) triggers the variation of the germination, development and ecophysiological behaviour of *L. pedunculata*, two microcosm assays were carried out in a growth chamber under controlled conditions (temperature of 20 ± 1 °C and photoperiod of 16 h light/8 h dark). All the seeds were disinfected, with 5% sodium hypochlorite for five minutes under agitation at 300 rpm, and then washed with abundant distilled water.

Both assays were set up according to the following combinations (each one with four replicates): CS-LC, CS-L, US-LC and US-L.

#### 4.2.1. Germination Assay

A germination microcosm assay was set up in pots with 25 seeds of LC and L and ≈200 g of soil (per pot). The pots were kept at 80% of the water holding capacity of the soil and the germinated seeds were registered every two days for four weeks. The criterion of germination was the emergence of a radicle through the seed coat.

#### 4.2.2. Plant Development and Ecophysiological Assay

Seeds from both sources (LC and L) were germinated on moist filter paper in Petri dishes and irrigated with distilled water until individuals reached a height of ≈1 cm and a root length of ≈0.5 cm. Seedlings (25 per pot) were transplanted to different pots (*n* = 4) containing ≈2 kg of each soil. Pots were kept at 80% of the water holding capacity of the soil in a growth chamber under controlled conditions for 35 days. Each five days, the height of the plants and the length and width of the leaves were measured. Leaf area was obtained by the values of length and width of five primary leaves from four individuals selected randomly in each pot and by applying the equations of Nakamura et al. [49].

At the end of the assay (35 days after seedlings transplantation), the plants were harvested, and the shoots were separated from the roots. The height and fresh biomass of each individual, as well as length and width of the leaves were determined. Roots and shoots were washed with tap water followed by distilled water. Additionally, roots were also washed in an ultrasound bath for 30 min. Subsamples of roots and shoots were frozen in liquid nitrogen and kept at −80 °C in a deep-freezer to determine physiological parameters as concentration of pigments, H_2_O_2_ and antioxidant metabolites, as well as enzyme activities. Remaining subsamples were dried at 40 °C and finely ground for multielemental analysis.

### 4.3. Analytical Methods

#### 4.3.1. Soil Characterization

Soils were characterized for pH determined in a water suspension (1:2.5 *m*/*v*), organic C [50], extractable P and K (Egner-Riehm method) (LV ST ZM 82-97) and total N (Kjeldahl method).

The multielemental concentration of the soils in the total fraction was determined by ICP-OES and INAA, after acid digestion with HClO_4_ + HNO_3_ + HCl + HF (Activation Laboratories). The element concentration of the available soil fraction was determined by ICP-OES after extraction using the rhizosphere-based method [51].

#### 4.3.2. Multielemental Concentration in Shoots and Roots of *L. pedunculata* and Plant Physiological Analysis

Samples of shoots and roots (LC and L) were digested with H_2_O_2_ (99%) and ultrapure concentrated HNO_3_ (69%) in a microwave oven [52,53], and the extracts were analysed by ICP-OES. Certified reference samples of bush branches and leaves (NCSDC73348) and blanks were used in parallel to test the accuracy of the method.

All plant physiological analyses were carried out with frozen leaves and roots (≈0.1 g) ground in liquid nitrogen. All analyses were done by spectrophotometry in a microplate reader (Sinergy HT, Biotec, Winooski, VT, USA).

In order to quantify pigment contents (chlorophylls, anthocyanins and carotenoids) frozen leave samples were macerated in acetone with Tris-HCl 100 mM (80:20). Chlorophyll *a* (chl *a*), chlorophyll *b* (chl *b*), total chlorophyll (chl t), anthocyanins and carotenoids contents were determined at 537, 647 e, 663 nm (for chlorophyll and anthocyanins) and 470 nm (carotenoids) using the concentrations obtained by the equations described by Richardson et al. [54] and Sims and Gamon [55].

Hydrogen peroxide production was determined by means of the fluorometric horseradish peroxidase (HRP) linked assay (Amplex Red assay kit, Invitrogen) following the methodology described by Creissen et al. [56]. The absorbance was recorded at 570 nm.

Ascorbate contents (ascorbic (AsA) and dehydroascorbic (DAsA) acids) were assayed according to the method proposed by Carvalho and Amâncio [57] adapted from Okamura [58]. Standard curves of AsA in the range of 10–60 mM were prepared in 5% metaphosphoric acid. The concentration of DAsA was calculated from the total ascorbate assayed subtracting the AsA concentration. All measurements were performed at 525 nm.

Glutathione contents, reduced (GSH) and oxidized (GSSG), were determined by colourimetry by the 2-vinyl pyridine method [59]. All measurements were performed at 412 nm. The reduction ratio (GSH percentage in the total glutathione pool) is defined as GSH/(GSH + GSSG) × 100.

### 4.4. Data Analysis

Statistical analyses were performed using the statistical software SPSS version 23.0 for Windows. Data were evaluated for normality (Shapiro-Wilk test) and homogeneity of variances (Levene’s test) and, when possible, a simple ANOVA and Tukey test (*p* < 0.05) were applied. For statistical purposes, the results below the detection limit were assumed as half of the detection limit. Data not satisfying these assumptions were analysed using a non-parametric analysis of the Kruskal-Wallis test (*p* < 0.05) and the Man-Whitney U Test for comparison. Quality control of soil and plant analysis was made by using technical replicates (three or four depending on the analysis), certified standards solutions, blanks and reference plant samples as well as quality standards of the international certified laboratory at the Activation Laboratories (ISO/IEC 17025) (total concentration in soils).

Leaf area ratio (LAR) was calculated from the leaf area and dry biomass by dividing the measured leaf area by the quantified dry biomass. This parameter is defined as the leaf area (in m^2^) that is used to produce one gram of dry biomass.

The translocation coefficient (TranslC = [total element in shoots]/[total element in roots]) was calculated in order to characterize the translocation capacity of an element from roots to leaves [60]. The soil-plant transfer coefficient (TransferC = [total shoots element]/[total soil element]) was calculated in order to characterize the accumulation behaviour [61]. Plants are considered as accumulators with TransferC > 1 or excluders/non-accumulators with TransferC < 1 of an element [62].

## 5. Conclusions

The assays were carried out with *L. pedunculata* from seeds of two provenances, São Domingos and Caldeirão, grown in soils from both locations. The soil from São Domingos had unfavourable conditions to enable plant growth, such as low pH and high total concentrations of several PHE that exceeded the intervention and maximum permitted levels for the protection of ecosystems and human health.

*Lavandula pedunculata,* regardless of the seed origin, activated defence mechanisms against oxidative stress caused by high concentrations of metal(loid)s. Plants grown from seeds collected in non-contaminated areas showed a high adaptation to extreme conditions. This ability makes *L. pedunculata* an adequate species to be used in phytostabilization programmes in areas with soils contaminated with elements such as As, Cr, Cu and Pb. Furthermore, this species showed a greater growth capacity when seeds from a contaminated area were sown in uncontaminated soils. Therefore, *L. pedunculata* already growing in mine soils can also be used as a species to improve the vegetation cover in uncontaminated soils under the influence of a Mediterranean climate.

## Figures and Tables

**Figure 1 plants-11-00105-f001:**
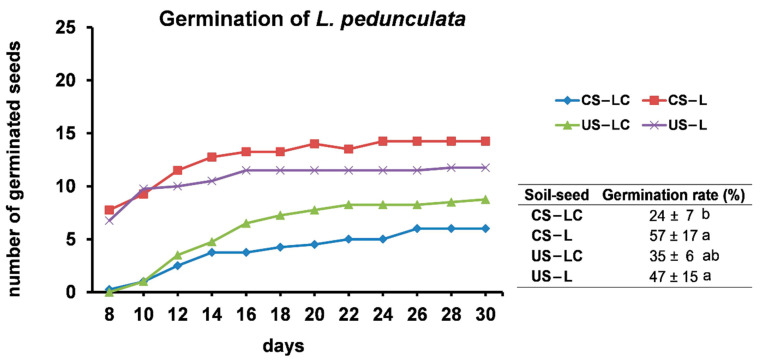
Germination of *L. pedunculata* seeds from São Domingos (LC) and Caldeirão (L) growing in the contaminated soil from São Domingos (CS) and the uncontaminated soil from Caldeirão (US) over time (30 days). Inset table shows the values of the germination rate after 30 days (average ± standard error, n_seeds_ = 25; n_pots_ = 4). Different letters indicate significant differences among soil/plant treatments for germination rates at 30 days (*p* < 0.05).

**Figure 2 plants-11-00105-f002:**
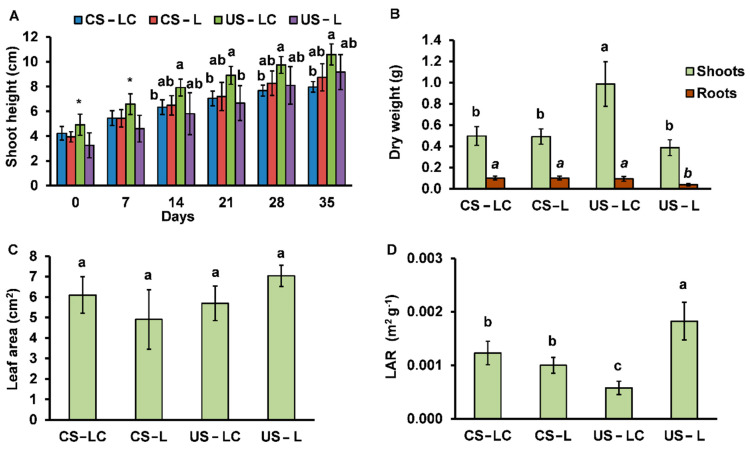
Shoot height (**A**), dry biomass (**B**), leaf area (**C**) and leaf area ratio (LAR) (**D**) of *L. pedunculata* (grown from São Domingos seeds (LC) and Caldeirão seeds (L)) planted in the contaminated soil from São Domingos (CS) and the uncontaminated soil from Caldeirão (US). (average ± standard error, *n* = 4). Different letters indicate significant differences among soil/plant treatments (*p* < 0.05). * indicates no significant differences among plant/soil combinations (*p* < 0.05).

**Figure 3 plants-11-00105-f003:**
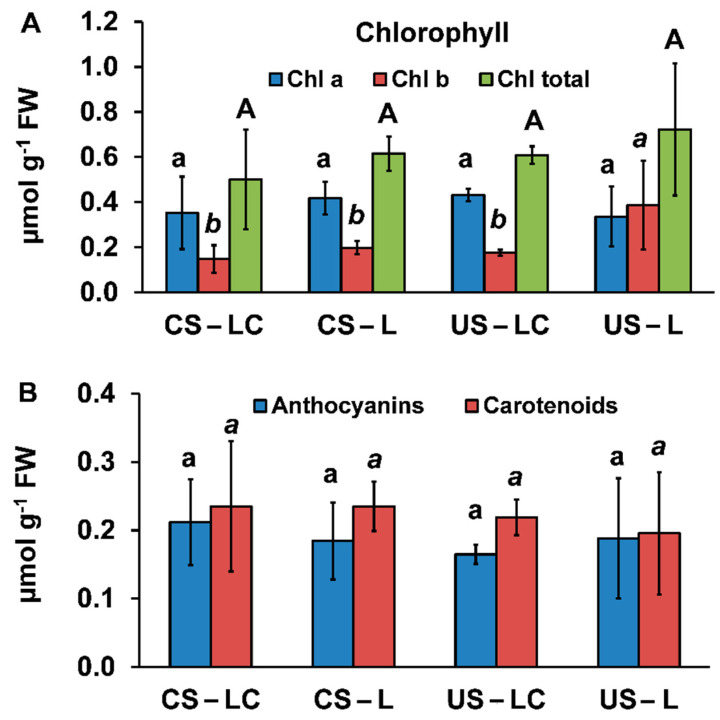
(**A**) Chlorophyll *a* (Chl *a*), chlorophyll *b* (Chl *b*) and total chlorophyll (Chl total) contents and (**B**) anthocyanins and carotenoids contents in shoots of *L. pedunculata* (from São Domingos seeds (LC) and Caldeirão seeds (L)) planted in the contaminated soil from São Domingos (CS) and the uncontaminated soil from Caldeirão (US) (average ± standard error; *n* = 4). Different letters—lowercase (Chl *a* and anthocyanins), italic (Chl *b* and carotenoids) and uppercase (Chl total)—indicate significant differences among soil/plant treatments (*p* < 0.05).

**Figure 4 plants-11-00105-f004:**
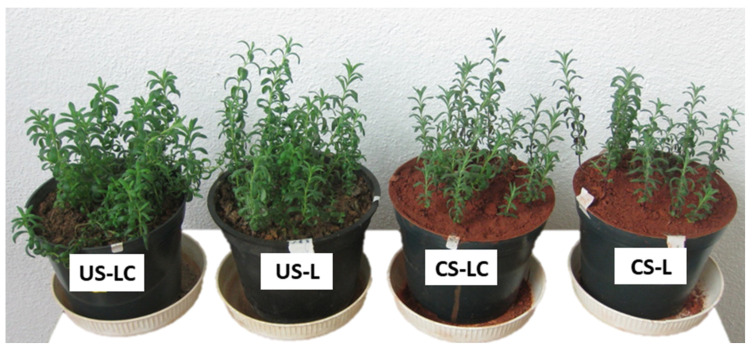
Aspect of soil-plant sets (US-LC, US-L, CS-LC, CS-L) after 35 days growing. (US: contaminated soil from São Domingos. US: uncontaminated soil from Caldeirão. L: seeds from Caldeirão area (non-contaminated). LC: seeds from São Domingos area (contaminated).

**Figure 5 plants-11-00105-f005:**
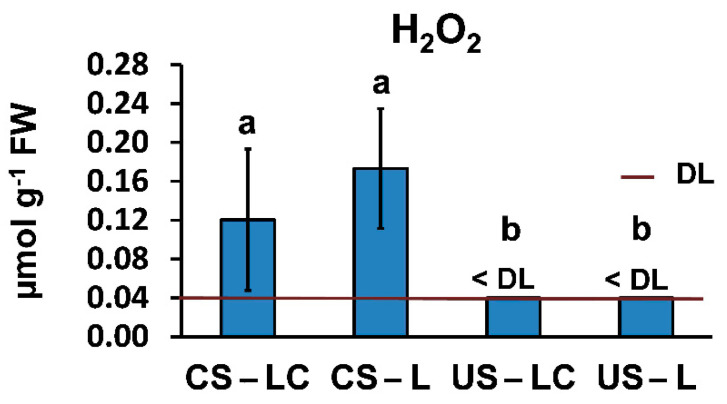
Hydrogen peroxide (H_2_O_2_) contents in shoots of *L. pedunculata* grown from São Domingos seeds (LC) and Caldeirão seeds (L) planted in the contaminated soil from São Domingos (CS) and the uncontaminated soil from Caldeirão (US) (average ± standard error; *n* = 4). Different letters indicate significant differences among soil/plant treatments (*p* < 0.05). DL: detection limit.

**Figure 6 plants-11-00105-f006:**
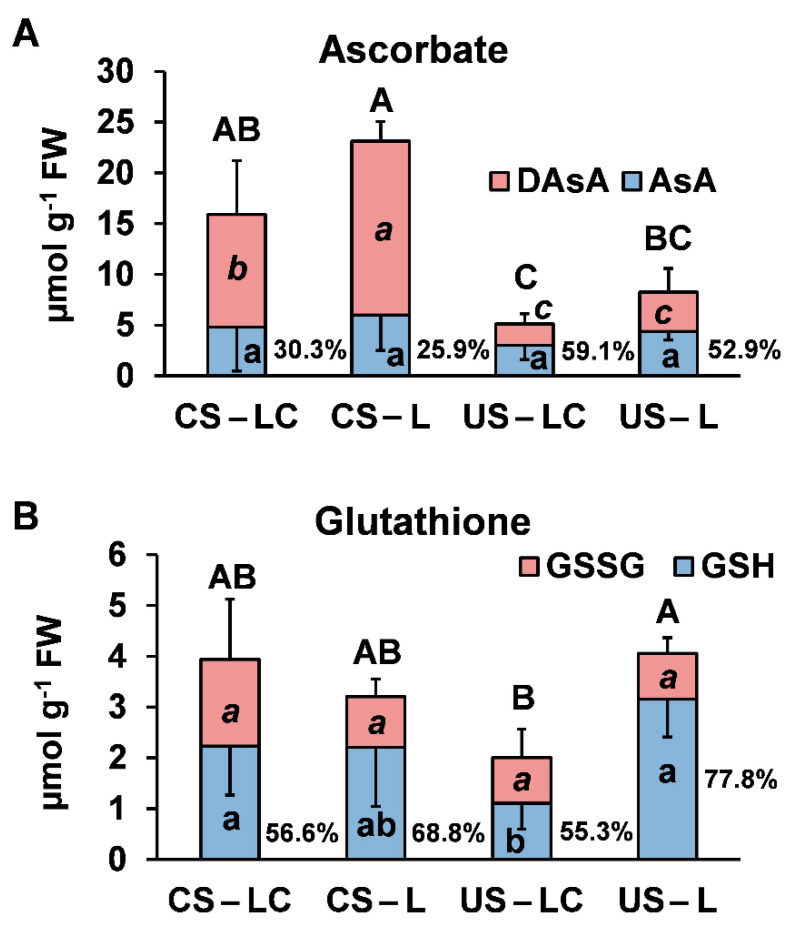
Reduced (AsA) and oxidized (DAsA) ascorbate contents (**A**) and reduced (GSH) and oxidized (GSSG) glutathione contents (**B**) in shoots of *L. pedunculata* grown from São Domingos seeds (LC) and Caldeirão seeds (L) planted in the contaminated soil from São Domingos (CS) and the uncontaminated soil from Caldeirão (US) (average ± standard error; *n* = 4). Percentage reduction of AsA and GSH are shown in each corresponding graph. Different letters—lowercase (AsA and GSH), italic (DAsA and GSSG) and uppercase (total ascorbate and total glutathione)—indicate significant differences among soil/plant treatments (*p* < 0.05).

**Table 1 plants-11-00105-t001:** Characteristics and multielemental total concentration (±standard error) in the contaminated soil from São Domingos (CS) and the uncontaminated soil from Caldeirão (US).

	Units	CS	US
pH_H2O_	-	3.51 ± 0.06 ^a^	5.54 ± 0.16 ^b^
Organic C	g kg^−1^	5.03 ± 0.25 ^a^	4.05 ± 0.24 ^b^
Total N_(Kjeldahl)_	g kg^−1^	0.16 ± 0.01 ^a^	0.15 ± 0.01 ^a^
C/N	-	31.44 ± 0.26 ^a^	27.00 ± 0.25 ^b^
Extractable P	mg kg^−1^	17.4 ± 0.03 ^a^	5.28 ± 1.48 ^b^
Extractable K	mg kg^−1^	459 ± 210 ^a^	153 ± 13 ^b^
		**Total Concentrations of Elements**	
As	mg kg^−1^	9993 * ± 1190 ^a^	18.7 * ± 0.9 ^b^
Ca	g kg^−1^	<0.1 ^b^	1.0 ± 0.0 ^a^
Cd	mg kg^−1^	0.9 ± 0.4 ^a^	0.9 ± 0.2 ^a^
Cr	mg kg^−1^	104 * ± 10 ^a^	<2 ^b^
Cu	mg kg^−1^	311 * ± 33 ^a^	61.3 ± 22.9 ^b^
Fe	g kg^−1^	236 ± 23 ^a^	33.7 ± 2.8 ^b^
K	g kg^−1^	3.3 ± 0.1 ^b^	18.9 ± 3.1 ^a^
Mg	g kg^−1^	<0.1 ^b^	3.9 ± 0.9 ^a^
Mn	mg kg^−1^	50.3 ± 7.0 ^b^	256 ± 175 ^a^
Mo	mg kg^−1^	11.0 ± 1.0 ^a^	<2 ^b^
Na	g kg^−1^	0.8 ± 0.1 ^b^	3.9 ± 0.2 ^a^
Ni	mg kg^−1^	16.0 ± 2.0 ^b^	42.7 ± 15.3 ^a^
Pb	g kg^−1^	477 * ± 168 ^a^	33.4 ± 1.6 ^b^
Sb	mg kg^−1^	2123 * ± 40 ^a^	1.6 ± 0.1 ^b^
Zn	mg kg^−1^	111 ± 10 ^a^	69.3 ± 25.1 ^b^
		**Elements’ Concentrations in the Soil Available Fraction**	
As	mg kg^−1^	2.41 ± 0.25 ^a^	0.34 ± 0.19 ^b^
Ca		573 ± 14 ^a^	297 ± 17 ^b^
Cd		0.08 ± 0.03 ^a^	<dl
Cr		0.01 ± 0.01 ^b^	0.12 ± 0.02 ^a^
Cu		0.05 ± 0.04 ^a^	<dl
Fe		31.2 ± 3.1 ^b^	80.0 ± 5.1 ^a^
K		8.86 ± 2.21 ^b^	52.8 ± 7.5 ^a^
Mg		75.3 ± 4.3 ^b^	107 ± 5 ^a^
Mn		9.35 ± 0.70 ^a^	6.4 ± 0.4 ^b^
Mo		0.13 ± 0.09 ^a^	<dl
Na		69.0 ± 6.7 ^a^	36.7 ± 2.4 ^b^
Ni		<dl	0.04 ± 0.01 ^a^
Pb		2.28 ± 0.65 ^a^	<0.01 ^b^
Sb		0.01 ± 0.06 ^a^	<dl
Zn		3.58 ± 1.08 ^a^	0.78 ± 0.48 ^b^

Values followed by a different letter indicate significance differences between metal contents in soils (*p* < 0.05). <dl: under detection limit. Potentially hazardous elements: * Total concentrations above the intervention and maximum permitted levels according to CCME (2007).

**Table 2 plants-11-00105-t002:** Concentrations of metal(oid)s in roots and shoots of *L. pedunculata* from seeds collected in São Domingos (LC) and Caldeirão (L) planted in the contaminated soil from São Domingos (CS) and the uncontaminated soil from Caldeirão (US) (*n* = 4).

Element	Units	CS-LC	CS-L	US-LC	US-L
		Roots			
As	mg kg^−1^	416 ± 104 ^a^	292 ± 110 ^a^	23.6 ± 11.9 ^b^	100 ± 70 ^b^
Ca	g kg^−1^	7.87 ± 4.51 ^a^	8.35 ± 1.16 ^a^	5.64 ± 1.78 ^b^	6.30 ± 2.31 ^ab^
Cd	mg kg^−1^	32.3 ± 10.0 ^a^	28.3 ± 12.8 ^a^	<0.3 ^b^	<0.3 ^b^
Cr	mg kg^−1^	10.1 ± 4.8 ^a^	7.66 ± 5.7 ^ab^	6.05 ± 2.56 ^b^	<2 ^c^
Cu	mg kg^−1^	60.1 ± 12.3 ^a^	45.4 ± 13.2 ^ab^	29.5 ± 12.5 ^b^	32.2 ± 21.6 ^b^
Fe	g kg^−1^	5.64 ± 1.31 ^a^	3.64 ± 1.38 ^b^	0.98 ± 0.46 ^c^	1.21 ± 0.17 ^c^
K	g kg^−1^	5.22 ± 1.12 ^c^	4.56 ± 0.67 ^c^	9.86 ± 3.06 ^b^	15.26 ± 6.10 ^a^
Mg	g kg^−1^	6.31 ± 1.47 ^a^	5.37 ± 1.20 ^a^	2.52 ± 0.83 ^b^	3.16 ± 1.14 ^b^
Mn	mg kg^−1^	106 ± 19 ^a^	110 ± 23 ^a^	60.8 ± 30.0 ^b^	87.5 ± 33.7 ^ab^
Mo	mg kg^−1^	4.82 ± 2.67 ^a^	6.07 ± 2.25 ^a^	<1 ^b^	<1 ^b^
Na	g kg^−1^	12.1 ± 2.0 ^a^	10.3 ± 3.2 ^a^	2.13 ± 0.41 ^b^	3.28 ± 1.48 ^b^
Ni	mg kg^−1^	<1 ^c^	<1 ^c^	38.6 ± 28.7 ^b^	152 ± 65 ^a^
Pb	mg kg^−1^	907 ± 215 ^a^	603 ± 213 ^a^	<3 ^b^	<3 ^b^
Sb	mg kg^−1^	<0.1 ^a^	<0.1 ^a^	<0.1 ^a^	<0.1 ^a^
Zn	mg kg^−1^	330 ± 51 ^a^	485 ± 164 ^a^	175 ± 92 ^b^	268 ± 105 ^ab^
**Element**	**Units**	**Shoots**			
As	mg kg^−1^	41.1 ± 21.6 ^a^	50.8 ± 18.4 ^a^	9.71 ± 5.66 ^b^	13.3 ± 9.2 ^b^
Ca	g kg^−1^	6.40 ± 1.53 ^a^	8.14 ± 2.48 ^a^	6.25 ± 2.17 ^a^	7.68 ± 0.58 ^a^
Cd	mg kg^−1^	<0.3 ^a^	<0.3 ^a^	<0.3 ^a^	<0.3 ^a^
Cr	mg kg^−1^	<2 ^a^	<2 ^a^	<2 ^a^	<2 ^a^
Cu	mg kg^−1^	4.12 ± 2.15 ^b^	4.10 ± 2.10 ^b^	11.6 ± 1.5 ^a^	10.7 ± 4.1 ^a^
Fe	g kg^−1^	0.47 ± 0.23 ^c^	0.71 ± 0.30 ^bc^	0.78 ± 0.27 ^b^	1.05 ± 0.14 ^a^
K	g kg^−1^	11.1 ± 3.8 ^b^	12.3 ± 4.6 ^b^	39.3 ± 12.6 ^a^	42.1 ± 3.7 ^a^
Mg	g kg^−1^	5.58 ± 1.31 ^a^	6.15 ± 2.50 ^a^	3.37 ± 1.33 ^a^	5.32 ± 1.44 ^a^
Mn	mg kg^−1^	564 ± 164 ^a^	522 ± 197 ^a^	478 ± 161 ^a^	525 ± 169 ^a^
Mo	mg kg^−1^	<1 ^b^	6.01 ± 3.11 ^a^	<1 ^b^	<1 ^b^
Na	g kg^−1^	4.68 ± 0.93 ^a^	6.43 ± 2.93 ^a^	2.61 ± 0.95 ^b^	2.85 ± 0.43 ^b^
Ni	mg kg^−1^	6.15 ± 3.42 ^ab^	31.8 ± 19.5 ^a^	3.48 ± 1.82 ^b^	5.12 ± 1.62 ^ab^
Pb	mg kg^−1^	53.2 ± 32.45 ^a^	60.6 ± 20.0 ^a^	<3 ^b^	<3 ^b^
Sb	mg kg^−1^	<0.1 ^a^	<0.1 ^a^	<0.1 ^a^	<0.1 ^a^
Zn	mg kg^−1^	179 ± 40 ^b^	196 ± 68 ^b^	228 ± 76 ^b^	337 ± 18 ^a^

Values for each parameter followed by a different letter are significantly different (*p* < 0.05).

**Table 3 plants-11-00105-t003:** Translocation coefficient (TranslC) and Soil-plant shoot transfer coefficient (TransferC) of metal(loid)s in *L. pedunculata* from São Domingos (LC) and Caldeirão (L) growing in the contaminated soil from São Domingos (CS) and the uncontaminated soil from Caldeirão (US) (values correspond to average; *n* = 4).

Element	CS-LC	CS-L	US-LC	US-L
	TranslC			
As	0.10	0.17	0.41	0.13
Ca	0.81	0.97	1.11	1.22
Cd	*	*	*	*
Cr	*	*	*	*
Cu	0.07	0.09	0.39	0.33
Fe	0.08	0.20	0.80	0.87
K	2.13	2.70	3.99	2.76
Mg	0.88	1.15	1.34	1.68
Mn	5.32	4.75	7.86	6.00
Mo	*	0.99	*	*
Na	0.39	0.62	1.23	0.87
Ni	*	*	0.09	0.03
Pb	0.06	0.10	*	*
Sb	*		*	*
Zn	0.54	0.40	1.30	1.26
	**TransferC**			
As	<0.01	0.01	0.52	0.71
Ca	64.00	81.40	6.25	7.68
Cd	*	*	*	*
Cr	*	*	*	*
Cu	0.01	0.01	0.19	0.17
Fe	<0.01	<0.01	0.02	0.03
K	3.36	3.73	2.08	2.23
Mg	55.80	61.50	0.86	1.36
Mn	11.21	10.38	1.87	2.05
Mo	*	0.55	*	*
Na	5.85	8.04	0.67	0.73
Ni	0.38	1.99	0.08	0.12
Pb	1.59	1.81	*	*
Sb	*	*	*	*
Zn	1.61	1.77	3.29	4.86

* Values were not calculated because contents in shoots and/or roots were under the detection limit (Table 2).

## Data Availability

All data used in and created by this study are included in this publication as tables and figures.

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
