# Peer review of "Influence of Seed Source and Soil Contamination on Ecophysiological Responses of Lavandula pedunculata in Rehabilitation of Mining Areas"

_plants, 2021, doi:10.3390/plants11010105_

Round 1
Reviewer 1 Report
Comments on the MS entitled " Influence of Seed Source and Soil Contamination on Ecophysiological Responses of Lavandula pedunculata in Rehabilitation of Mining Areas´ bearing the MS no plants-1500866 authored by Lago et al.
Although the authors have made sincere efforts to prove that applicability of Lavandula pedunculata In rehabilitation of mining contaminated soil without any methodological flaws, however there is little novelty. Such studies with phytoremediation ability of wild plants have been reported widely in the literature, with very little applicability. Nonetheless, the study is worth of publication after reviewing in view of the concerns
Abstract: It is lengthy lacking focus. Just mention the objective, bit of methodology and the significant outcome/s without any discussion.
Introduction does not elaborate about the geographical conditions for the eco-habitat of Lavandula pedunculata and its significance both economic as well as ecological.
The experimentation has a significant drawback as the authors have not compared the phytoextraction efficiency with already estabilished plants i.e. Vetiver, Brassica etc.
Conclusion: is lengthy without the mentioning the significant findings. The conclusion should be precise and definitive without any discussions.
Few corrections have been made in the pdf copy attached, however these is not all.
The MS may be accepted after Minor review

Author Response
Comments on the MS entitled " Influence of Seed Source and Soil Contamination on Ecophysiological Responses of Lavandula pedunculata in Rehabilitation of Mining Areas´ bearing the MS no plants-1500866 authored by Lago et al.
Although the authors have made sincere efforts to prove that applicability of Lavandula pedunculata In rehabilitation of mining contaminated soil without any methodological flaws, however there is little novelty. Such studies with phytoremediation ability of wild plants have been reported widely in the literature, with very little applicability. Nonetheless, the study is worth of publication after reviewing in view of the concerns
Abstract: It is lengthy lacking focus. Just mention the objective, bit of methodology and the significant outcome/s without any discussion.
Done, the abstract was adapted according to your suggestions
Introduction does not elaborate about the geographical conditions for the eco-habitat of Lavandula pedunculata and its significance both economic as well as ecological.
Thank you very much for your suggestion but in the section 4 (M&M) we indicate these geographical conditions:
“I) São Domingos mining area is an old metal-sulfide mine abandoned since 1960, where mining activities began in the pre-roman period to extract Ag, Au and Cu from gossan. Subsequently, in the second half of the nineteenth century and the first half of the twentieth century, massive sulphides, mainly pyrite, were processed to obtain Cu, Zn, Pb and S, among other elements [42]. Mining activities produced large amounts of wastes, which were spread and deposited in several tailings along the area, with different environmental risk according to their volume, chemical and mineralogical characteristics, and potential generation of acid mine drainage [43]. The low pH of the mine wastes and their leachates and the high mobility of the potentially toxic elements (PHE) generated extreme environmental conditions and severe contamination in the entire mine and the surrounding areas, including surface water [43], [44]. In fact, soils developed on mine wastes and host rocks (Spolic Technosol; [45]) as well as many adjacent natural soils, which are developed on schists and greywackes (Lithic Leptosols; [45]), present high total concentrations of metal(loil)s [46].
- II) The climate in this area is semiarid and mesothermic (Thornthwaite classification), with low rainfall (annual precipitation of 548 mm, Climate normal 1981–2010 from Beja station [47]) and below to the potential evapotranspiration, with warm winters and hot summers within continental climate regimes.
The Serra do Caldeirão is the largest mountain in the Algarve region, which marks the border between the Algarve and Alentejo regions. The lithology of Serra do Caldeirão is mainly composed by schists and greywackes rocks, which originated incipient and shal-low soils (Lithic Leptosols and Hyperskeletic/Skeletic Leptosols [45]). It constitutes a physical barrier for cold north-easterly winds, humid southerly winds and north-west depressions, contributing to the existence of a Mediterranean climate with weak annual rainfall around ≈1000 mm and mild winter temperatures due to the altitude [48].”
Besides, in the introduction section we mention the economic and ecological significance of Lavandula:
“Currently, the studies on the applicability of L. pedunculata are mainly focused on characterising and evaluating the bioactivity of essential oils and other extracts of this species with antioxidant, antifungal and antibacterial properties [17], [20], [21], [22], [23], [24]. Lavandula pedunculata shows high levels of free-radical scavengers and therefore it can be a potential source of active metabolites with a positive effect on human health. Hence, the antioxidant properties of this species might be an economic asset for popula-tions if it was used for soil recovery in contaminated areas [9], [19].
The experimentation has a significant drawback as the authors have not compared the phytoextraction efficiency with already estabilished plants i.e. Vetiver, Brassica etc.
The authors consider that the objective of this study is not to stablish comparisons with another species, we are not choosing the species with the best or with no phytoextraction efficiency. Anyway, we include several references of previous papers where we indicate the results obtained in similar previous studies.
Conclusion: is lengthy without the mentioning the significant findings. The conclusion should be precise and definitive without any discussions.
We have simplified and reorganized the conclusions.
Few corrections have been made in the pdf copy attached, however these is not all.
Thank you, they have been taken into account
The MS may be accepted after Minor review
Reviewer 2 Report
Nowadays,Mining activities have become many areas of the Iberian Pyrite Belt (IPB) into extreme environments with high concentrations of metal(loid)s. These harsh conditions could inhibit or reduce colonization and/or development of most vegetation. but, some species or populations have developed ecophysiological responses to tolerate these stress factors and contaminated soils. In order to stduy which ecological factor (source of the reproductive plant material or soil conditions) triggers the variation of the development and physiological behaviour of Lavandula pedunculata (Mill.) Cav., two microcosm assays were carried out with seeds and soils collected in two locations to study this effect. The results from this manuscript are new and quite signicant and the paper is well-written. I only have following two minor points:
Minor Comments:
- For the Figure 1 and 2, it is rather difficult to understand that US-LC had longer shoot length and shoot dry weight, why the leaf area is smaller than the other treatment? Furthermore, CS-LC had better germination rate than US-LC, why it had shorter shoot length and lower shoot dry weight ?
- For the data in each column of Table 1 and Table 2,it should have same decimal point to make all the data consistent.
Author Response
Nowadays,mining activities have become many areas of the Iberian Pyrite Belt (IPB) into extreme environments with high concentrations of metal(loid)s. These harsh conditions could inhibit or reduce colonization and/or development of most vegetation. but, some species or populations have developed ecophysiological responses to tolerate these stress factors and contaminated soils. In order to stduy which ecological factor (source of the reproductive plant material or soil conditions) triggers the variation of the development and physiological behaviour of Lavandula pedunculata (Mill.) Cav., two microcosm assays were carried out with seeds and soils collected in two locations to study this effect. The results from this manuscript are new and quite signicant and the paper is well-written. I only have following two minor points:
Minor Comments:
- For the Figure 1 and 2, it is rather difficult to understand that US-LC had longer shoot length and shoot dry weight, why the leaf area is smaller than the other treatment? Furthermore, CS-LC had better germination rate than US-LC, why it had shorter shoot length and lower shoot dry weight?
The shoot height in US-LC is significantly higher than CS-LC, which is due to the fact that the edaphological characteristics of the CS soil (São Domingos) are much more unfavourable than those of the US soil (Caldeirao). Regarding the leaf area, there are no significant differences between the different microcosms. In any case, LAR (ratio: leaf area / dry leaf mass) indicates that -without significant differences in the leaf area- leaf mass is higher for LC in CS (contaminated soil) than in US (uncontaminated soil). This shows that there is a greater development capacity of LC individuals when they are not subjected to the oxidative stress of the Sao Domingos soil (CS).
Regarding the Germination rate, LC in the US soil (not contaminated) shows greater germination under better stress conditions and also a higher height and dry weight in the shoots than those in CS-LC (As shown in Figures 2A and 2B). This is due to the greater growth capacity of the LCs (coming from the contaminated site - São Domingos) when they are planted in a soil with lower concentrations of metal (loid) s and better soil characteristics.
- For the data in each column of Table 1 and Table 2,it should have same decimal point to make all the data consistent.
Done, thank you.
Reviewer 3 Report
The manuscript "Influencce of Seed Source and Soil contamination on Ecophysiological Responses of Lavandula Penduculata in Rehabilitation of Mining Areas" presented for evaluation to Plants MDPI Journal is interesting in terms of its of its relevance to practice and pretty well well-written. The described research concerns one of the species from the Lamiaceae family growing in the area of the Iberian pyrite Belt (Spolic Technosol) and the reference material from a metal-free area (Lithic Leptosols). Physico-chemical characteristics of the contaminated (CS) and uncontaminated (US) soils were checked. Seed samples were taken to start an experimental set-up. The methodology applied does not raise any reservations. After germination assay plant development were studied on different soil and plant material were, in a way, cross-tested. The concentrations were determined in soils (total and available fractions) and plants (shoots and roots). The parameters (shoot height, dry biomass, leaf area and leaf area ratio) associated to plant development was taken. It was ascertained that the total concentrations of As, Cr, Cu, Pb and Sb in CS and As in US exceed the maximum limits for ecosystem protection and human health. Authors found that the LC plants showed higher growth and shoot biomass production and lower root/shoot dry biomass in US than in CS. The L plants growing in CS showed a rapid adaptation to the mine soil conditions, with almost no symptom of toxicity and development and growth similar to LC plants. On the basis of biochemical analysis they concluded among other that Lavandula pedunculata, regardless of its origin, has the ability to adapt to the extreme conditions of mine soils.
However, the authors did not avoid mistakes and inaccuracies.
- In the title, the genre epithet should be written with a lowercase letter: therefore not "Lavandula Penduculata" but there should be: Lavandula penduculata
- After the summary and the keywords, there must be an explanation of the abbreviations contained in the text
- The introduction should be more in line with the methodological assumptions of the experiment. Currently, it is written in a way that is too vague , which does not introduce the reader exactly to the issues presented by the authors in the methodology and results.
- The References section is to be thoroughly improved in terms of the spelling of species names (I checked in the quoted texts 4, 5, 6, 7, 8, 9, 10, 13, 14, 15, 20, 22, 28, 36, 41, etc. genre names in italics), in lines 594-599 and 655-658, please correct the entry.
Author Response
The manuscript "Influencce of Seed Source and Soil contamination on Ecophysiological Responses of Lavandula Penduculata in Rehabilitation of Mining Areas" presented for evaluation to Plants MDPI Journal is interesting in terms of its of its relevance to practice and pretty well well-written. The described research concerns one of the species from the Lamiaceae family growing in the area of the Iberian pyrite Belt (Spolic Technosol) and the reference material from a metal-free area (Lithic Leptosols). Physico-chemical characteristics of the contaminated (CS) and uncontaminated (US) soils were checked. Seed samples were taken to start an experimental set-up. The methodology applied does not raise any reservations. After germination assay plant development were studied on different soil and plant material were, in a way, cross-tested. The concentrations were determined in soils (total and available fractions) and plants (shoots and roots). The parameters (shoot height, dry biomass, leaf area and leaf area ratio) associated to plant development was taken. It was ascertained that the total concentrations of As, Cr, Cu, Pb and Sb in CS and As in US exceed the maximum limits for ecosystem protection and human health. Authors found that the LC plants showed higher growth and shoot biomass production and lower root/shoot dry biomass in US than in CS. The L plants growing in CS showed a rapid adaptation to the mine soil conditions, with almost no symptom of toxicity and development and growth similar to LC plants. On the basis of biochemical analysis they concluded among other that Lavandula pedunculata, regardless of its origin, has the ability to adapt to the extreme conditions of mine soils.
However, the authors did not avoid mistakes and inaccuracies.
- In the title, the genre epithet should be written with a lowercase letter: therefore not "Lavandula Penduculata" but there should be: Lavandula penduculata
Done, thank you
- After the summary and the keywords, there must be an explanation of the abbreviations contained in the text
Done, thank you for your suggestion.
- The introduction should be more in line with the methodological assumptions of the experiment. Currently, it is written in a way that is too vague, which does not introduce the reader exactly to the issues presented by the authors in the methodology and results.
We have included various remarks in the introduction to make the reader more familiar with the issues exposed in the methodology and results
- The References section is to be thoroughly improved in terms of the spelling of species names (I checked in the quoted texts 4, 5, 6, 7, 8, 9, 10, 13, 14, 15, 20, 22, 28, 36, 41, etc. genre names in italics), in lines 594-599 and 655-658, please correct the entry.
Done, thank you.